# Reducing the Bitter Taste of Pharmaceuticals Using Cell-Based Identification of Bitter-Masking Compounds

**DOI:** 10.3390/ph15030317

**Published:** 2022-03-07

**Authors:** Leopoldo Raul Beltrán, Sonja Sterneder, Ayse Hasural, Susanne Paetz, Joachim Hans, Jakob Peter Ley, Veronika Somoza

**Affiliations:** 1Department of Physiological Chemistry, University of Vienna, 1090 Vienna, Austria; leopoldo_beltran@outlook.de (L.R.B.); sonja.sterneder@univie.ac.at (S.S.); a01347723@unet.univie.ac.at (A.H.); 2Symrise AG, Ingredient Research Flavor & Nutrition, 37603 Holzminden, Germany; susanne.paetz@symrise.com (S.P.); joachim.hans@symrise.com (J.H.); jakob.ley@symrise.com (J.P.L.); 3Leibniz-Institute of Food Systems Biology at the Technical University of Munich, 85354 Freising, Germany; 4Nutritional Systems Biology, Technical University of Munich, 85354 Freising, Germany

**Keywords:** bitter taste, bitter-masking, cell assay, sensory analysis, palatability of pharmaceuticals

## Abstract

The palatability of a pharmaceutical preparation is a significant obstacle in developing a patient-friendly dosage form. Bitter taste is an important factor for patients in (i) selecting a certain drug from generic products available in the market and (ii) adhering to a therapeutic regimen. The various methods developed for identification of bitter tasting and bitter-taste modulating compounds present a number of limitations, ranging from limited sensitivity to lack of close correlations with sensory data. In this study, we demonstrate a fluorescence-based assay, analyzing the bitter receptor TAS2R-linked intracellular pH (pH_i_) of human gastric parietal (HGT-1) cells as a suitable tool for the identification of bitter tasting and bitter-taste modulating pharmaceutical compounds and preparations, which resembles bitter taste perception. Among the fluorometric protocols established to analyze pH_i_ changes, one of the most commonly employed assays is based on the use of the pH-sensitive dye SNARF-1 AM. This methodology presents some limitations; over time, the assay shows a relatively low signal amplitude and sensitivity. Here, the SNARF-1 AM methodology was optimized. The identified bicarbonate extrusion mechanisms were partially inhibited, and measurements were carried out in a medium with lower intrinsic fluorescence, with no need for controlling external CO_2_ levels. We applied the assay for the screening of flavonoids as potential bitter-masking compounds for guaifenesin, a bitter-tasting antitussive drug. Our findings revealed that eriodictyol, hesperitin and phyllodulcin were the most potent suitable candidates for bitter-masking activity, verified in a human sensory trial.

## 1. Introduction

Many drugs taste bitter and are thus often aversive to children and adults. Reducing the bitter taste of pharmaceutical preparations is widely accepted as a way of improving patient adherence and compliance to a therapeutic drug regimen, particularly when considering pediatric and geriatric populations.

For adults, encapsulation of a medicine in the form of a pill or a tablet may be an effective method to reduce its unpleasant bitter taste. Children are not only very aversive to bitter tastants [1], which would require the use of larger amounts of encapsulation material, but also many of them cannot or will not swallow solid dosage forms. Another strategy to reduce the bitterness of drugs is through additions of bitter-taste masking compounds, such as sugars, acids, salts or flavor compounds [2,3,4]. Although pleasant flavorings may help children consume some medicines, they often are not effective in fully suppressing bitter taste perceptions and do not guarantee improved palatability, which is a main driver of compliance to drug therapy.

In attempts to develop palatable pharmaceutical preparations, several methods have been established to measure bitter taste perception in animal models and in humans by means of “electronic tongues” [5,6], or using computational approaches targeted at predicting bitterness from a chemical structure [7]. These methods have limitations. Although human sensory studies representatively reveal the bitter taste, toxicity data are needed prior to administering these tests. Moreover, relatively large amounts of pharmaceutical preparations are required to be tested by a sensory panel. Results from animal studies are limited by species-dependent expression of bitter receptors [8], resulting in bitter responses differing from those of humans. The electronic tongue system uses sensors and may provide a first clue about feasibilities of taste-masking strategies in formulation development, and allows comparison of already existing products. Nevertheless, correlation of its results with human sensory data is limited for many formulations, most likely because physiological conditions are not taken into account [9]. This also holds true for computational approaches, which, in addition, do not allow researchers to identify bitter-masking compounds which may act antagonistically by allosteric or competitive inhibition of a given bitter receptor [7].

In one of our own previous studies, we developed an in vitro cell model for the identification of bitter tasting and bitter-taste modulating food constituents [10]. This cell model is based on human gastric parietal tumor (HGT-1) cells, which have been demonstrated to express all bitter taste receptors (TAS2Rs) located on the tongue, and respond to TAS2R activation by changes in proton secretion [11]. One commonly used agent for this noninvasive measurement of intracellular pH is carboxy-seminaphthorhodafluor-1 (carboxy-SNARF-1) [12]. This compound presents several advantages over classical fluorochromes such as 2,3-dicyano-hydrochinone (DCH) or 2,7-bis(2-carboxyethyl)-5(6)-carboxyfluorescein (BCECF). Its low leakage and larger shift in its emission spectrum as a function of pH provide the ability to resolve small differences in pH over longer periods of time [13]. However, this fluorescence-based test system has been developed for the identification of bitter tasting and/or bitter-taste modulating food constituents and is, as such, not suitable for pharmacological preparations. The test system’s relatively high signal-to-noise ratios and low amplitude signals require comparatively large amounts of receptor ligands, and limit applications to ligands with its low fluorescent activity. This presented research demonstrates adaptation of the human gastric parietal cell-based bitter test to pharmaceutical preparations by taking into account activation patterns of human TAS2Rs in a native cell system and validation of in vitro results by means of a human sensory trial.

Gastric parietal cells are mainly located in the glands of the fundus mucosa. They are pyramidal cells each with a large basal surface next to the basement membrane and an apex towards the lumen of the gland. There is a complex system of intracellular canaliculi within their cytoplasm; a high number of mitochondria and intracellular vesicles containing H^+^/K^+^-ATPases are the main characteristics of these cells [14]. With these characteristics, parietal cells specialize in the energy-intensive task of counter-gradient electrolyte transport. They secrete high amounts of H^+^ and Cl^−^ across their apical membranes and are able to form a gradient of more than four million-fold across the membranes of their secretory canaliculi. Even at rest, the lumen of the stomach can have a pH of less than 4 [15]. Upon stimulation, the complex tubulovesicle structure in which the H^+^/K^+^-ATPases are located tends to disappear as it fuses with the secretory canaliculi, thus considerably expanding cell surfaces [14,16]. This translocation of proton pumps from the cytosol to the secretory surface increases the amount of protons that can be secreted. However, for each proton that is secreted, an equal amount of OH^−^ is left in the cytoplasm. This OH^−^ is rapidly converted by carbonic anhydrases into HCO_3_^−^ which is subsequently exported by basolateral ion exchangers, thereby preventing the detrimental effects of excessive intracellular alkalinization. Key players in this process are anion exchanger proteins, such as the HCO_3_^−^/Cl^−^ exchanger AE2 and the Na^+^/H^+^ exchanger [17].

The HGT-1 cell line originates from primary cells of a gastric adenocarcinoma, established by Laboisse et al. in 1982 [18]. HGT-1 cells present epithelial morphology and are able to form microvilli and tight junctions. In addition, they have been shown to express the principal ion transporters involved in acid secretion, e.g., H^+^/K^+^-ATPase, AE2 and a Na^+^/H^+^ exchanger thought to correspond to the NHE4 isoform detected in parietal cells [18,19].

HGT-1 cells provide a valuable model for the study of proton secretion, and, due to the bitter receptors they express, they have also been used as a screening tool for bitter-taste modulating food constituents [11]. Prior to the studies of Liszt et al. [11], several fluorometric studies have used HGT-1 cells in order to investigate changes at the intracellular pH level induced by bitter-tasting food constituents [10]. Based on these findings, HGT-1 cells have been validated to significantly correlate between their proton secretory activity in vitro and changes in the intra-gastric pH in vivo, induced by bitter tasting compounds and bitter taste modulators [10,11].

However, the HGT-1 cell line, as well as almost every proton-secreting cell line, presents the unique challenge that, while it is specialized in secreting protons, it also produces high amounts of intracellular HCO_3_^−^ (Figure 1) which must be quickly removed from the cells. This rapidly corrects any pH imbalance, which in turn limits fluorometric information that could be obtained. In addition to this, early observations on the ultrastructural changes related to the functional activity of parietal cells [16] showed that most of the proton pumps are not located at the cell membrane, but at the vesicle reservoir level; the high mobilization of vesicles containing H^+^/K^+^-ATPases consequently increases the cell surface after agonist stimulation. Therefore, recorded signal amplitudes may on occasion be too low. As a consequence of all these factors, the standard methodology for evaluating changes in the intracellular pH of HGT-1 cells has presented several limitations: its relatively low amplitude signals, and the fact that relatively high amounts of agonists are required in order to observe a marked change in proton secretion [10,11]. Moreover, high signal-to-noise ratios and the need to precisely control external CO_2_ levels in order to prevent long time alkalization of the extracellular pH have been observed, particularly when using the standard extracellular medium DMEM which relies on the HCO_3_^−^/CO_2_ buffer system for regulating extracellular pH. In one of our recent works, we used KREBS-Ringer HEPES buffer instead of the cell culture medium DMEM [17] in order to directly compare results from a serotonin-release assay with those of the proton secretion assay in HGT-1 cells [20].

With this research, we present an in vitro assaying method for the determination of bitter tasting and bitter taste-masking pharmaceuticals. For this purpose, we optimized an already existing protocol using human parietal cell line HGT-1 cells. The suitability of this method was demonstrated by means of a known bitter tasting drug, guaifenesin, and the reduction in the guaifenesin-mediated bitter response by addition of flavonoids as bitter-masking compounds. Finally, we verified our findings with a human sensory study. Specifically, the sensitivity of the assay was enhanced by (i) removal of HCO_3_^−^ via adaptation of the cell culture medium, and (ii) the usage of inhibitors of solute carrier (SLC) transporters. The improved methodology was validated by studying the effects of histamine as one of the most potent endogenous stimulants of proton secretion, and omeprazole that potently inhibits proton secretion by inhibiting H^+^/K^+^-ATPase in the secretory membranes of parietal cells [21,22,23]. The suitability of the HGT-1 cell model to identify compounds that potently mask the bitterness perceived from the pharmaceutical, bitter tasting compound guaifenesin was demonstrated and validated by a human sensory trial. A graphical overview of the experimental approach is shown in Appendix A.

## 2. Results

### 2.1. Adaptation of Cell Culture Medium: DMEM vs. Krebs-HEPES Buffer

The standard proton secretion-assay for HGT-1 cells uses DMEM as an extracellular medium during measurements. Although an optimal medium for cell viability, DMEM presents two disadvantages: it has a relatively high intrinsic fluorescence, and it requires adequate control of external CO_2_ levels. As a first step to optimize the proton secretion assay, we investigated whether the non-CO_2_/HCO_3_^−^ dependent and comparatively inexpensive Krebs-HEPES buffer (KRHP) could be used instead, and whether it would present a better signal-to-noise ratio. For this purpose, the signal-to-noise ratio from SNARF-measurements performed under KRHP and under DMEM were compared. It was observed that cells under KRHP presented a nearly ten times higher signal-to-noise ratio than their counterparts under DMEM after 10 min of treatment [10,11] (Figure 2a). In addition, we observed that, under atmospheric CO_2_ levels, vehicle-treated HGT-1 cells under KRHP presented significantly less change in intracellular pH after 35 min as compared with vehicle-treated cells under DMEM (Figure 2b).

Finally, we compared both media in a 30-min long measurement, where recordings under KRHP presented a more consistent profile over long time, in addition to reduced variability (Figure 3).

### 2.2. Improving Assay Sensitivity by Means of Inhibitors of SLC Transporters and Pre-Treatment with Histamine

Parietal cells must quickly remove the HCO_3_^−^ excess produced as a consequence of proton secretion. This task is commonly in charge of members of the solute carrier (SLC) family of ion exchangers. Since HCO_3_^−^ removal results in a weaker signal from the pH-sensitive dye SNARF, our next step consisted in applying inhibitors of two well-characterized SLC transporters: Anion Exchanger 2 (AE2) and Na^+^/H^+^ exchanger 1 (NHE1). In accordance with this, we decided to partially inhibit those exchangers with the following blockers: 4,4′-Diisothiocyanatostilbene-2,2′-disulfonic acid disodium (DIDS) for AE2, and 5-(N-Ethyl-N-isopropyl) amiloride (EIPA) for NHE1. Both compounds were applied at a concentration of 20 µM. Since most H^+^/K^+^-ATPases are located in tubulovesicle organelles [24] and only mobilize after agonist exposure, HGT-1 cells were pre-treated with 10 µM of the standard agonist histamine. Under this treatment, marked changes in HGT-1 cell surfaces were observed (Appendix A). In detail, a change in the morphology from the inactivated state to the stimulated state with an increased cell surface area was demonstrated. This indicates a higher amount of H^+^/K^+^-ATPases at the cell membrane level.

As previously stated, one potential limitation from the standard SNARF-methodology on HGT-1 cells is its relatively low sensitivity. This is exemplified by the case of histamine, for which concentrations of up to 1 mM are commonly employed [10,11]. We asked whether the modifications we designed could increase system sensitivity to histamine concentrations one order of magnitude lower than those commonly used. Figure 4 shows the effect of histamine at concentrations of 100 and 1000 µM over a period of 30 min on HGT-1 cells under simultaneous application of EIPA [20 µM] and DIDS [20 µM]. As a result, histamine-pretreated cells showed higher signal amplitudes and a more stable response to 100 µM histamine. Overall, applying these changes rendered the system more stable over time in comparison with the previously employed methodology (Figure 4).

The next step consisted of investigating whether the histamine response could be inhibited by micromolar concentrations of the proton pump inhibitor omeprazole. Here, it was demonstrated that co-application of 100 µM histamine and with 10 µM omeprazole abolished the histamine-induced change (Figure 5).

Finally, in order to determine a confidence interval for each measurement, we built a control range for the distribution of IPX values for vehicle solution-treated cells (Figure 6). Based on these data, a 95% confidence interval was calculated, which ranged for the values of the intracellular proton index (IPX) from −0.2 to 0.17.

### 2.3. Identification of Flavonoids That Mask the Bitter Taste of Guaifenesin

Guaifenesin is a broadly used expectorant with a reportedly bitter taste [25]. Since an approximate concentration of 10 mM guaifenesin is expected to be reached in the stomach’s gastric juice after administration of a tablet containing 600 mg guaifenesin with a glass of water, HGT-1 cells were treated with a guaifenesin concentration of 10 mM, following the presently described optimized protocol. As a result, the guaifenesin-treated HGT-1 cells demonstrated a reduced IPX value compared to non-treated and to 100 µM histamine (Figure 7).

Since prior research on various flavonoids demonstrated a reducing effect on the IPX response induced by bitter-tasting food constituents [10,11], HGT-1 cells were co-treated with guaifenesin and the flavonoids eriodictyol, homoeriodictyol, hesperitin or phyllodulcin. With the exception of phyllodulcin, each of the flavonoids co-applied at a concentration of 30 µM reduced the bitter response of 10 mM guaifenesin over time (Figure 8).

### 2.4. Sensory Evaluation of the Bitter-Masking Effect of Eriodictyol, Homoeriodictyol Hesperitin and Phyllodulcin on the Bitter Taste of Guaifenesin

As a next step, a panel of 15 to 31 trained volunteers evaluated the bitter perception of guaifenesin and its reduction by co-administration of homoeriodictyol, eriodictyol, hesperitin or phyllodulcin. For this purpose, the panel was trained with in-water solutions of 500 ppm caffeine and 5 ppm quinine hydrochloride serving as bitter references. After completion of training, the panelists were asked to rank the bitterness of guaifenesin (dissolved in water at 1333 to 13,333 ppm equal to 6.7 to 67.3 mM, Figure 9) by sip and spit administration on a structured, continuous scale of 0 (no bitter taste) to 10 (very strong bitter taste).

The highest concentration of guaifenesin at 13,333 ppm was chosen for the sensory trial as it is a commonly used concentration in liquid cough formulations. As a result, all concentrations of guaifenesin administered were evaluated to be significantly more bitter than water. Afterwards, the panelists were asked to rank changes in the bitterness perceived by 1333 ppm (6.7 mM) guaifenesin when co-administered with 100 ppm (330.8 µM hesperitin); 100 ppm (346.9 µM) eriodictyol; 7 ppm (24.5 µM) phyllodulcin; or 100 ppm (331 µM) homoeriodictyol. Among the tested flavonoids, eriodictyol and hesperitin were able to reduce the perceived bitterness of guaifenesin (1333 ppm) in a manner that resembled the effects of these compounds on the changes induced by guaifenesin on the IPX index of HGT1 cells at minute 0 (Figure 10). The bitter-masking effect by phyllodulcin did not reach the level of statistical significance (*p* = 0.105), but demonstrated a trend toward the respective result in the HGT-1 cell assay.

## 3. Discussion

A central challenge of administering oral medicine to patients, especially to children, is the matter of taste, since many drugs taste unpleasant, with bitterness being the primary culprit [26]. In other words, reducing the bitter taste of drugs is widely accepted as a way of improving patient adherence and compliance to a therapeutic regimen. The major current strategies to reduce the bitterness of drugs involve encapsulation and the addition of bitter-masking compounds. Both strategies have limitations; for encapsulation, additional solids that increase the size of the formulation to be swallowed are needed; the addition of bitter-masking compounds first require test systems in order to identify such compounds and to test their efficacy. Sensory studies are undoubtedly the best tools for the identification of bitter-tasting and bitter-modulating compounds and for evaluating their efficacy. However, toxicity data and the relatively large amounts of material needed limit their suitability as a screening tool.

Here, we introduce an in vitro screening model based on the fluorometric detection of bitter receptor-linked proton secretion by human gastric parietal cells in culture (HGT-1 cells). The HGT-1 cell line has been previously established as a screening model for the identification of bitter tasting and bitter-taste modulating food constituents [10,11]. This method has been established for testing food constituents in food-representative, comparatively large concentrations for which low sensitivity was reported as a minor limitation. In these studies, the endogenous gastric acid stimulant histamine is used as reference compound in concentrations two orders of magnitude higher than its saturating concentrations in physiologically relevant heterologous systems [10,11,27,28]. For the screening of drugs administered at lower doses, a higher sensitivity is needed not only for test compounds, but also for the reference compound histamine to be tested in physiologically relevant concentrations.

The present study has addressed this major issue of sensitivity and now demonstrates a cell model that is suitable for the identification of bitter-tasting and bitter-taste modulating pharmaceuticals. By using a medium that does not rely on a HCO_3_^−^/CO_2_ buffer system, test compounds can be measured for periods longer than 30 min, up to 60 min without devices externally controlling the CO_2_ concentrations. Compared to the commonly used DMEM cell culture medium, the use of a Krebs Ringer HEPES (KRHP) buffer also presents the advantage of resulting in a high signal-to-noise ratio, while also providing considerably more consistency over time. In addition, KRHP does not contain fetal bovine serum (FBS) proteins, which can potentially react with compounds under investigation via protein-compound interactions.

Another measure to improve sensitivity of the HGT-1 assay was to partially inhibit the bicarbonate extrusion mechanism by using the SLC inhibitors EIPA and DIDS, while simultaneously increasing the presence of H^+^/K^+^-ATPases at the cell membrane using a histamine pre-treatment. Mechanistically, the inhibitors of the AE2 and NHE1 SLC carriers minimize the diffusion of ions across the secondary active transport, thereby shifting the H^+^ transport towards the H^+^/K^+^-ATPase. Overall, these changes rendered the system responsive to histamine concentrations one order of magnitude less than those commonly employed, thereby expanding the usability of this assay as a pharmacological tool. The assay was then applied in order to identify compounds that effectively mask the bitterness of guaifenesin.

Guaifenesin is mainly described as an antitussive oral expectorant drug against excessive mucus accumulation in the respiratory tract [29]. In our study, guaifenesin at a concentration of 10 mM, a concentration reached in the stomach after the oral intake of cough medicine, stimulated proton secretion in the HGT-1 cell assay, indicating a bitter response. A bitter perception range of 1333–13,333 ppm (6.7–67.3 mM) guaifenesin was also demonstrated in the sensory trial, thereby confirming results from the HGT-1 cell assay. Whereas in the sensory trial, concentrations of 1333 ppm and 2667 ppm (6.7 µM and 13.4 µM, respectively) were perceived as more bitter than water, a comparable concentration of 10 µM (1990 ppm) guaifenesin reduced the IPX in the HGT-1 cell assay, thereby indicating a bitter response. Since Thawabteh et al. 2019 showed that guaifenesin activates the broadly tuned TAS2R14 [30], an involvement of this bitter receptor in the HGT-1 cell response and in bitter sensory perception seems likely. This hypothesis is supported by findings from Kagan et al. 2009 [31] who reported that only oral administration, but neither intraperitoneal nor intravenous injection of guaifenesin, stimulated respiratory secretion; furthermore, this response was not seen after abdominal surgery. The authors hypothesized that the action of guaifenesin on respiratory tract secretion was mediated by stimulation of the gastro-intestinal tract. With current knowledge, including the fact that guaifenesin targets at least one of the 25 human bitter receptors [30] and that TAS2Rs are involved in airway smooth muscle function [32,33], we hypothesize that guaifenesin elicits the HGT-1 response and the perception of bitter taste via TAS2R activation.

Next, we demonstrated the bitter response revealed for guaifenesin in the HGT-1 cell assay was ameliorated by the flavonoid compounds hesperitin, eriodictyol, homoeriodictyol and phyllodulcin, with hesperitin and phyllodulcin being most potent.

Concentrations of hesperitin, eriodictyol and homoeriodictyol, ranging from 7 ppm to 100 ppm, were chosen according to previously published sensory results pertaining to their bitter-masking properties on caffeine [34]. Phyllodulcin, however, though structurally similar to hesperitin, has not been described for its bitter-masking effects; it has been shown to elicit sweet taste perception while tasting bitter at higher concentrations of 75 ppm [35,36]. For eriodictyol and homoeriodictyol, reduction in the bitter response evoked by caffeine in the HGT-1 cell model has been shown for concentrations of 3, 30 and 300 µM [10]. In this study, using the more sensitive HGT-1 cell model, concentrations of eriodictyol and homoeriodictyol of one magnitude lower were applied and used as a reference for the test concentrations of hesperitin and phyllodulcin.

As a result, all of these flavonoids tested reduced the guaifenesin-evoked bitter response in the HGT-1 model, which is well in line with results from the sensory trial; the exception was with phyllodulcin for which the level of statistical significance was not reached. We hypothesize that the HGT-1 cell model is more sensitive than sensory trials. Sensory perception in the oral cavity is not solely determined by chemoreceptor activation, as is the case for the activation of chemoreceptors in the HGT-1 cell model. In the oral cavity, the oral microbiome and the composition of saliva might impact ligand-receptor activations [37]. Moreover, co-localization of human sweet and bitter taste receptors, as shown for solitary chemosensory cells in extra-oral tissues [38], may differ from those in HGT-1 cells, thereby accounting for differences in the bitter/sweet and bitter/sweet-masking responses.

From a mechanistic point of view, the question is whether bitter-masking compounds tested antagonistically target the same bitter taste receptor as guaifenesin agonistically acts on, which is the broadly tuned TAS2R14 [30]. For hesperitin, homoeriodictyol and eriodictyol, TAS2R14 has been reported as a target by Roland et al. [39]. In our study, homoeriodictyol did not mask the bitter response induced by guaifenesin, neither in the HGT-1 cell model, nor in the sensory trial. One reason for this discrepancy might be that homoeriodictyol does not solely target TAS2R14. A previous in vitro study using heterologous expression systems demonstrated agonistic and antagonistic activities of this compound towards different TAS2Rs [11]. That being said, the TAS2R activation pattern of a given compound or mixtures of a number of compounds determines the overall bitter or bitter-modulating response, thereby supporting the value of native cell systems or sensory trials to reveal sensory properties. Involvement of a given TAS2R in such a chemosensory response can be verified by means of CRISPR-Cas9 knock-out approaches which have been applied to HGT-1 cells already [11].

For the identification of potentially bitter or bitter-modulating pharmaceuticals and the development of palatable formulations of pharmaceuticals, the presented HGT-1 cell assay offers a valuable screening model. New compounds with unknown toxicology profiles or limited available quantities, in addition to compound combinations, can be screened on a large scale. Subsequently, the chemosensory activity of generally safe and promising compounds and/or combinations thereof can be validated by sensory panels.

## 4. Materials and Methods

### 4.1. Chemicals

For the cell culture experiments, the following chemicals were used: trypsin, glutamine, penicillin/streptomycin, DMEM and RPMI culture media; 4-(2-hydroxyethyl)-1-piperazineethanesulfonic acid (HEPES), dimethyl thiazolyl diphenyl tetrazolium (MTT) reagent, 5-(N-Ethyl-N-isopropyl) amiloride (EIPA), diisothiocyanatostilbene-2,2′-disulfonic acid disodium (DIDS), guaifenesin, histamine and omeprazole. Chemicals were acquired from Sigma-Aldrich. 1,5-carboxy-seminaphtorhodafluor acetoxymethyl ester (SNARF-1-AM), nigericin and fetal bovine serum were purchased from Invitrogen (Vienna, Austria). All other chemicals were obtained from Roth (Karlsruhe, Germany). Stock solutions of the evaluated compounds were prepared and stored at −20 °C for a period no longer than a week.

### 4.2. Cell Culture

HGT-1 cells were obtained from Dr. C. L. Laboisse (Inserm 94-04, Faculté de Medicine, Nantes, France, clone 6), and were cultured under standard tissue culture conditions at 37 °C and 5% CO_2_. The cell line was established in vitro from a primary gastric adenocarcinoma tumor of a 60-year-old male patient clone 6. Roswell Park Memorial Institute medium (RPMI) with glucose (4%, PAA, Coelbe, Germany) was used as culture medium and supplemented with 10% fetal bovine serum (PAA, Coelbe, Germany), 2% L-glutamine (PAA, Coelbe, Germany) and 1% penicillin streptomycin (PAA, Coelbe, Germany). Cells were cultivated until reaching 80% confluence and then passaged according to standard procedures.

### 4.3. Cell Viability

The impact of the evaluated compounds on cell viability was assessed using the tetrazole MTT (3-(4,5-dimethylthiazol-2-yl)-2,5-diphenyltetrazolium bromide) reagent. Cells were cultivated in a 96-well plate at a density of 10,000–100,000 viable cells per well and settled for 24–72 h. The assay was applied as previously explained [20]. Cell viability was determined relative to medium-only treated control cells (untreated controls = 100%).

### 4.4. Intracellular pH Measurement in HGT-1

For the determination of intracellular pH as a marker of proton secretion in HGT-1 cells, means of the pH-sensitive fluorescent dye SNARF-1-AM were quantified. HGT-1 cells were cultivated under standard conditions and seeded in a 96-well plate at a density of 10,000–100,000 viable cells per well. They were allowed to settle for 24–72 h, until 80% confluence was achieved. Afterward, fluorescent staining of the cells was performed with 3 μM SNARF-1-AM in KRHP (10 mM HEPES, 11.7 mM D-glucose, 4.7 mM KCl, 130 mM NaCl, 1.3 mM CaCl_2_, 1.2 mM MgSO_4_ and 1.2 mM KH_2_PO_4_, brought to a pH of 7.4 with 1 M NaOH) for 30 min at standard conditions either with or without histamine (10 μM), after one washing step with KRHP (100 µL/well). Subsequently, the cells were incubated with test substances in combination with 4,4′-Diisothiocyanatostilbene-2,2′-disulfonic acid disodium (DIDS, 20 µM) and 5-(N-Ethyl-N-isopropyl) amiloride (EIPA, 20 µM) after two washing steps with KRHP (100 µL/well). Histamine (100–1000 μM) served as positive control. Fluorescence was detected using an Infinite 200 Pro Plate Reader (Tecan, Männderdorf, Switzerland) at 580 and 640 nm emission after excitation at 488 nm. A ratio was gained using the emission wavelengths. The pH was calculated by a calibration curve generated by treating the cells with potassium buffer solutions (20 mM NaCl, 110 mM KCl, 1 mM CaCl_2_, 1 mM MgSO_4_, 18 mM D-glucose and 20 mM HEPES) of various pH values, ranging from 7.2 to 8.2 in the presence of 20 μM nigericin, to equilibrate intracellular and extracellular pH in the cells. The analyzed intracellular pH of the calibration solutions was fit to a linear regression and used for calculating the intracellular pH of the samples. The ratio between treated and medium-only/non-treated cells was formed and log2 transformed to determine the intracellular proton index (IPX). The lower the IPX, the stronger the proton secretion by the cell [10,11].

### 4.5. Sensory Analyses

A total of 31 trained panelists (mixed gender, ages between 18 and 65 years), with no history of known taste disorders or chronic medications, were recruited from Symrise AG (Holzminden, Germany). All assessors were informed about potential effects of the intended concentrations and exposures of guaifenesin during the study based on freely available information about dosages, risks and side effects validated by a medical doctor. All trials were performed by sip and spit tests; the concentrations of guaifenesin were chosen from average or lower dosages in cough relieves. The assessors provided informed consent to participate in the experiments conducted during the present study. Assessors were trained in sensory experiments with purified reference compounds for bitterness (caffeine at 500 ppm and quinine hydrochloride at 5 ppm in water). Guaifenesin, eriodictyol, homoeriodictyol, phyllodulcin and hesperitin were added to water solutions. All compounds were used in a purity higher than 95% according to quantitative HPLC or quantitative NMR analysis. Panelists were instructed to rinse their mouthes after each evaluation step with water and to wait for 2 min between each sample; a maximum of two tests were conducted in one sensory session. Ratings were carried out using EyeQuestion software on randomized samples with three-digit codes on a structured but continuous linear scale from 0 [no bitter taste] to 10 [very strong bitter taste]. Data were expressed as percentages of the linear range.

### 4.6. Statistical Analysis

Statistical analysis was performed using Excel 2016 (Microsoft) and R statistical language (R Foundation for Statistical Computing, Vienna, Austria). Outliers were excluded by Nalimov outlier analysis from both in vitro and sensory analysis. Effects on cellular viability of the test compounds on HGT-1 cells compared to non-treated cells were excluded by means of a two-tailed Student’s t-test and considered to be significant at *p* < 0.05. Significant differences in the data set of the intracellular pH were evaluated using a one-way ANOVA with Tukey’s range test for multiple comparisons. At least three biological replicates and two technical replicates were analyzed for each cell culture experiment. Data reported in the results section are stated as mean ± SEM, unless indicated otherwise.

## 5. Conclusions

This study presented an optimized in vitro HGT-1 cell assay which can be used as a valuable screening model for the identification of bitter tasting and bitter-modulating pharmaceuticals. It is especially suitable for compounds with unknown toxicological profiles or when available amounts of putatively bitter-tasting pharmaceuticals are limited.

## Figures and Tables

**Figure 1 pharmaceuticals-15-00317-f001:**
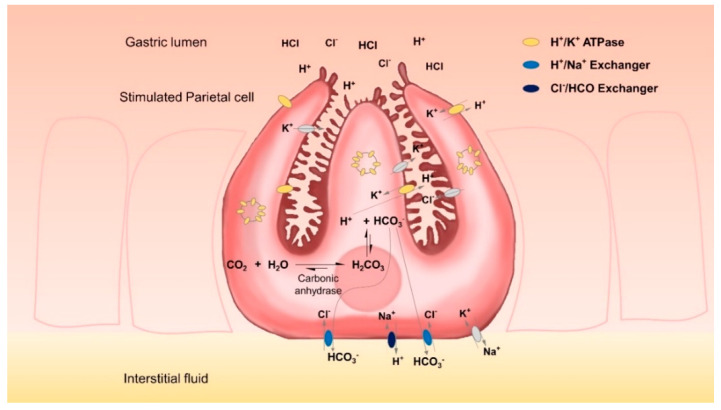
Schematic mechanism of proton secretion by HGT-1 cells. Carbonic anhydrases catalyze the reaction between CO_2_ and H_2_O to form carbonic acid, which immediately dissociates into H^+^ and HCO_3_^−^. H^+^ ions are secreted in exchange of K^+^ by H^+^/K^+^-ATPase antiporter pumps, of which the majority of these are located within the intracellular vesicles of unstimulated cells. In order to prevent intracellular alkalization, HCO_3_^−^ ions are extruded by several members of the solute carrier (SLC) transport proteins in exchange of highly abundant extracellular ions. In addition to this, other exchangers might create an alkalinized microenvironment at the basolateral membrane level in order to enhance the activity of these extrusion mechanisms.

**Figure 2 pharmaceuticals-15-00317-f002:**
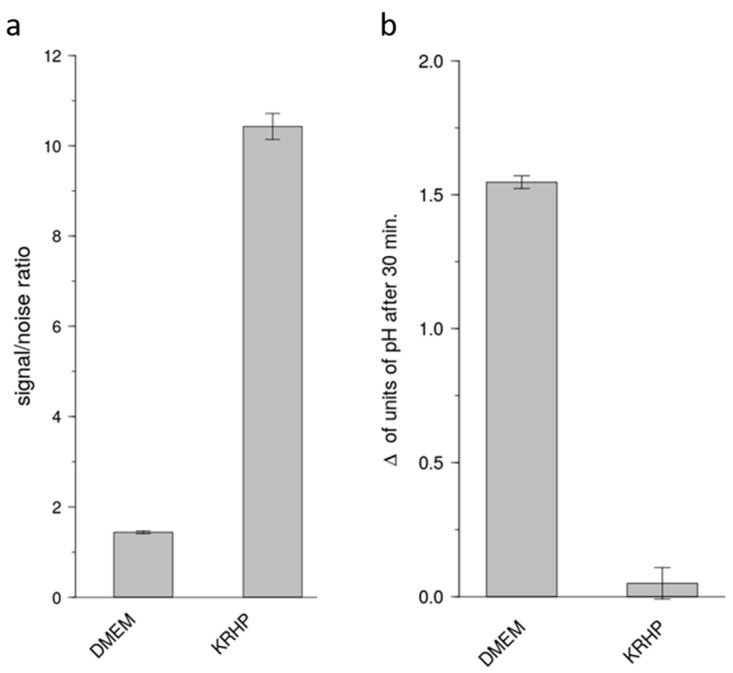
(**a**) Signal (SNARF-SNARF-1 AM loaded wells) to noise (blank wells) ratios of DMEM and KRHP media after 10 min of treatment. Data presented as mean ± SEM. Each group consisted of 96 wells from 5 independent experiments. (**b**) Difference in intracellular pH after 35 min exposure to vehicle solution. N of 5 biological replicates (BR).

**Figure 3 pharmaceuticals-15-00317-f003:**
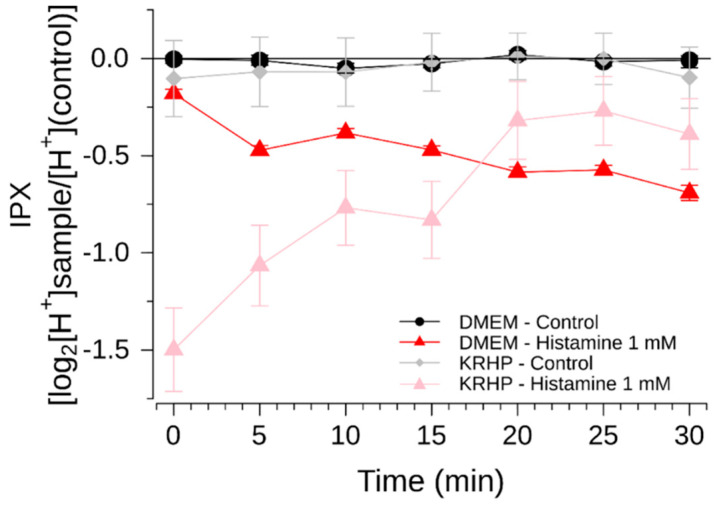
Intracellular proton index (IPX) of HGT-1 cells exposed to 1000 µM histamine under DMEM (red triangle) and under KRHP (pink triangle). Recordings were performed at room temperature at intervals of 5 min. Cells treated with vehicle solution served as negative control for both groups (gray diamonds and black circles). Data presented as mean ± SEM. N of 6 BR.

**Figure 4 pharmaceuticals-15-00317-f004:**
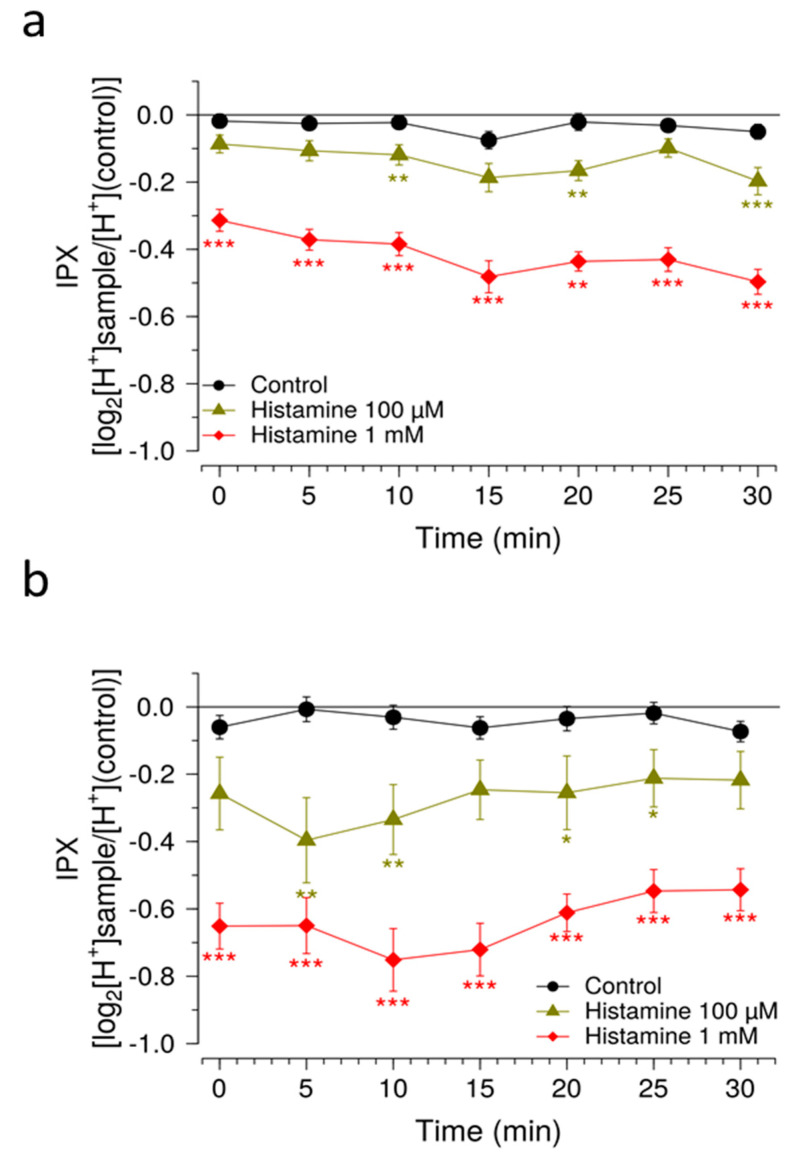
Histamine priming of HGT-1 cells. Intracellular proton index (IPX) of HGT-1 cells exposed to 100 and 1000 µM histamine (green diamonds and red squares, respectively) without (**a**) and with (**b**) 10 µM histamine pretreatment. Recordings were performed at room temperature at intervals of 5 min. Evaluated wells contained 20 µM EIPA and 20 µM DIDS for inhibiting HCO_3_^−^ extrusion systems. Cells treated with vehicle solution served as negative control (circles). Data presented as mean ± SEM. Statistical significance was evaluated using one-way ANOVA with Tukey’s range test for multiple comparisons. N of 4 BR, 8–12 TR. * denotes *p* < 0.05, ** denotes *p* < 0.01 and *** denotes *p* < 0.001 for the corresponding histamine concentration when compared to negative control.

**Figure 5 pharmaceuticals-15-00317-f005:**
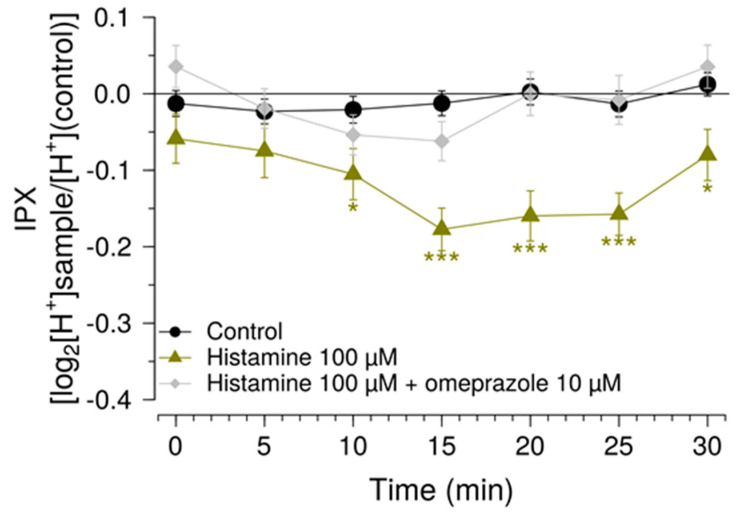
Effect of omeprazole on proton secretion by HGT-1 cells. Intracellular proton index (IPX) of HGT-1 cells exposed to 100 µM histamine alone (green triangles) or in combination with 10 µM omeprazole (grey diamonds). Recordings were performed at room temperature at intervals of 5 min. Cells treated with vehicle solution served as negative control (black circles). Data presented as mean ± SEM. Statistical significance was evaluated using one-way ANOVA with Tukey’s range test for multiple comparisons. * denotes *p* < 0.05 and *** denotes *p* < 0.001 for the corresponding compound, according to the assigned color when compared to the negative control. N of 6 BR, 12–24 TR.

**Figure 6 pharmaceuticals-15-00317-f006:**
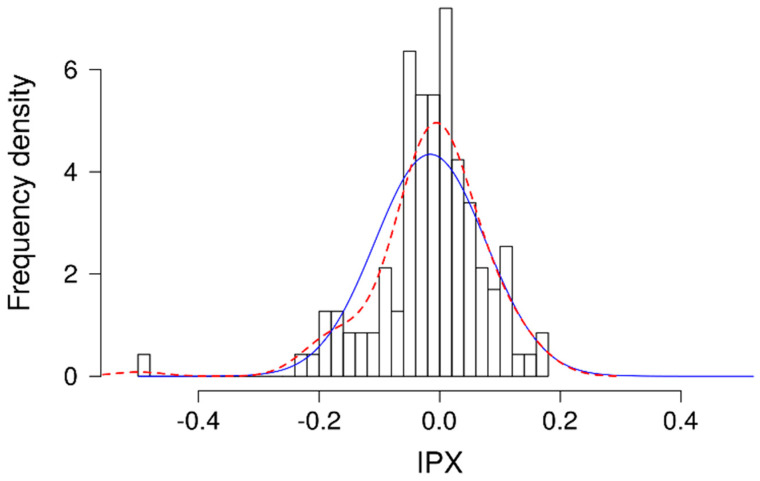
Distribution of IPX values for vehicle solution-treated cells. Density histogram for the intracellular proton index (IPX) of 118 technical replicates (control lanes). Data from 38 separate measurements. Mean of −0.015, SD of 0.092. The density curve for the data (red) is presented along a normal distribution curve with the above-mentioned parameters (blue). Based on these data, a 95% confidence interval was calculated, which ranged from −0.2 to 0.17. Plates in which more than one control lane presented an IPX average beyond the above-mentioned range were excluded from analysis. Similarly, plates in which more than 1 histamine control failed to induce an increase in proton secretion were also excluded.

**Figure 7 pharmaceuticals-15-00317-f007:**
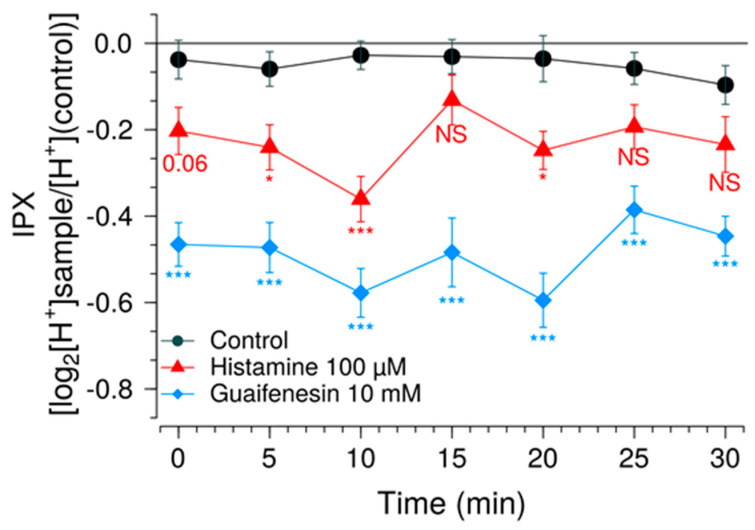
Effect of 10 mM guaifenesin on HGT-1 cells. Intracellular proton index (IPX) of HGT-1 cells exposed to 10 mM guaifenesin (blue diamonds). Recordings were performed at room temperature at intervals of 5 min. Histamine at a concentration of 100 µM (red triangles) was used as positive control. Cells treated with vehicle solution served as negative control (black circles). Data presented as mean ± SEM. Statistical significance was evaluated using one-way ANOVA with Tukey’s range test for multiple comparisons. * denotes *p* < 0.05 and *** denotes *p* < 0.001 for the corresponding compound according to the assigned color when compared to the negative control. N of 4 BR, 8–16 TR.

**Figure 8 pharmaceuticals-15-00317-f008:**
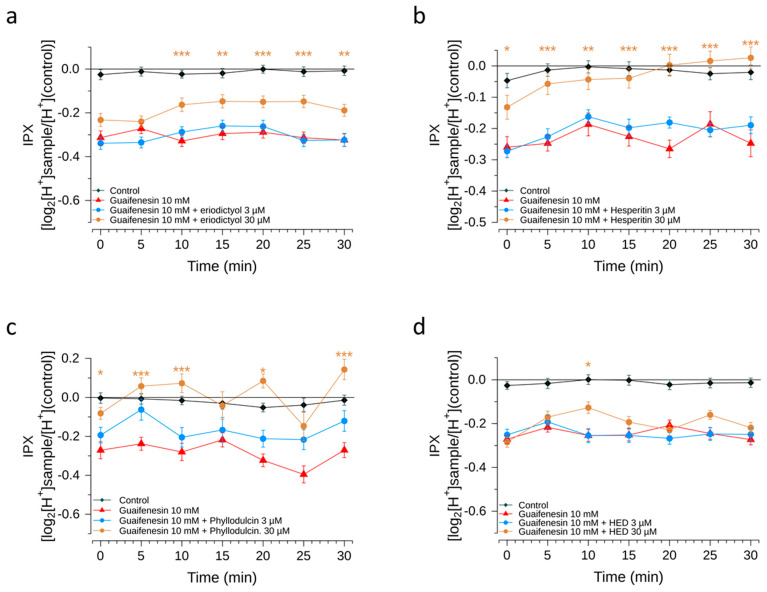
Guaifenesin modulation by bitter-masking compounds. Intracellular proton index (IPX) of HGT-1 cells exposed to 10 mM guaifenesin alone (red triangles) or in combination with of eriodictyol (**a**), hesperitin (**b**), phyllodulcin (**c**) and homoeriodictyol (**d**) at concentrations of 3 µM (blue circles) and 30 µM (brown circles). Recordings were performed at room temperature at intervals of 5 min. Cells treated with vehicle solution served as negative control (black diamonds). Data presented as mean ± SEM. Statistical significance was evaluated using one-way ANOVA with Tukey’s range test for multiple comparisons. * denotes *p* < 0.05, ** denotes *p* < 0.01 and *** denotes *p* < 0.001 for the corresponding compound according to the assigned color when compared to the negative control. N of 4 BR, 8–16 TR.

**Figure 9 pharmaceuticals-15-00317-f009:**
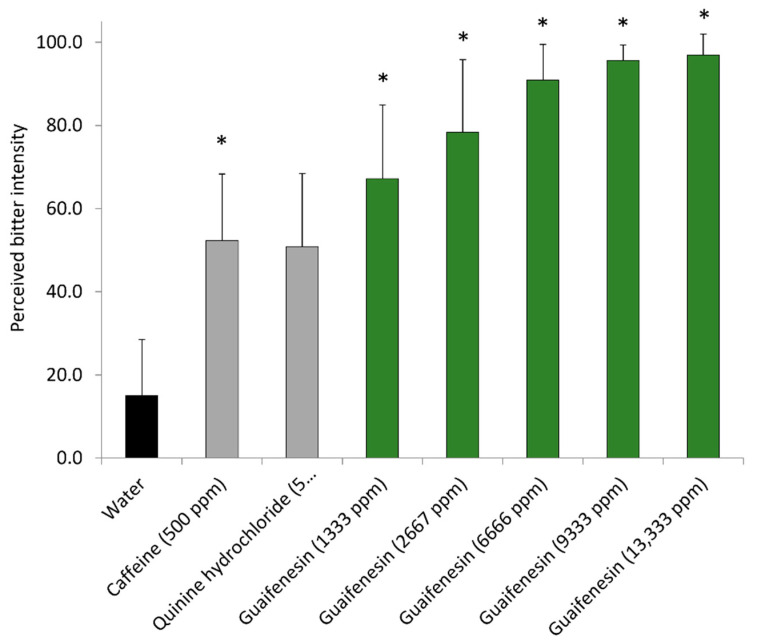
Bitter intensity of Guaifenesin. A panel of 15 trained volunteers were asked to taste (sip and spit) and rank the perceived bitterness (structured but continuous scale of 0 [no bitter taste]–100 [very strong taste]) of aqueous solutions containing typical concentrations of standard bitter compounds caffeine and quinine HCl and different concentrations of guaifenesin (1333 ppm–13,333 ppm, 6.7–67.3 mM) against water as control. Data presented as mean +/− standard deviation. Statistics: ANOVA on ranks with post-hoc Dunn’s method compared to control (water); * indicates significance with *p* ≤ 0.05.

**Figure 10 pharmaceuticals-15-00317-f010:**
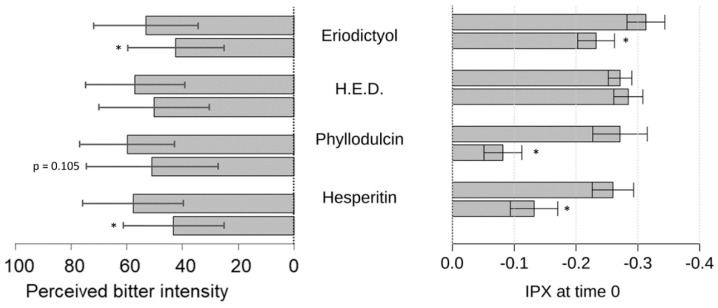
Modulating effect of eriodictyol, homoeriodictyol, phyllodulcin and hesperitin on guaifenesin bitterness and guaifenesin-induced changes in IPX index. (left) Guaifenesin 20 mg/15 mL (1333 ppm, 6.73 mM) was administered in combination with several bitter-masking compounds (eriodictyol (100 ppm), homoeriodictyol (100 ppm), phyllodulcin (7 ppm) and hesperitin (100 ppm)) and their mixtures to a panel of 25–31 trained volunteers. (right) IPX index of HGT-1 cells at the beginning of a 30-min measurement. Cells were exposed to 10 mM guaifenesin either alone or in combination with the indicated bitter-masking compound (eriodictyol (30 ppm), homoeriodictyol (30 ppm), phyllodulcin (30 ppm) and hesperitin (30 ppm). N of 3–5 biological replicates. All data presented as mean +/− sem. Statistical significance was evaluated using one-way ANOVA with Tukey’s range test for multiple comparisons. * denotes *p* < 0.05 for the corresponding compound when compared to the effect of guaifenesin.

## Data Availability

Not applicable.

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
