# Peer review of "Reducing the Bitter Taste of Pharmaceuticals Using Cell-Based Identification of Bitter-Masking Compounds"

_pharmaceuticals, 2022, doi:10.3390/ph15030317_

Round 1

Reviewer 1 Report

The present manuscript try to demonstrate a  fluorescence-based assay, analyzing the bitter receptor (TAS2R)- linked intracellular pH (pHi) of human gastric parietal (HGT-1) cells as a suitable tool for the identification of bitter tasting and bitter taste modulating pharmaceutical compounds and preparations, resembling the sensory bitter perception.

Some remarks to the authors:

page 1, line 5 I think that the word "and" is not necessary.

page 1, line 12 Did you mean: "The palatability of a pharmaceutical"?

page 5, line 170-172 "This section may be divided by subheadings. It should provide a concise and precise description of the experimental results, their interpretation, as well as the experimental conclusions that can be drawn." -this text must be deleted.

The article is very interesting and it can be published after a minor revision.

Author Response

Comments provided by Reviewer #1

The present manuscript try to demonstrate a  fluorescence-based assay, analyzing the bitter receptor (TAS2R)- linked intracellular pH (pHi) of human gastric parietal (HGT-1) cells as a suitable tool for the identification of bitter tasting and bitter taste modulating pharmaceutical compounds and preparations, resembling the sensory bitter perception.

Comment 1: page 1, line 5 I think that the word "and" is not necessary.

Response: We deleted the word “and” in line 5.

Comment 2: page 1, line 12 Did you mean: "The palatability of a pharmaceutical"?

Response: Thank you for your comment and apologies for having missed the preposition. We changed the sentence from “The palatability a pharmaceutical preparation is a significant obstacle in developing a patient-friendly dosage form.“ to “The palatability of a pharmaceutical preparation is a significant obstacle in developing a patient-friendly dosage form.“ in line 12.

Comment 3: page 5, lines 170-172 "This section may be divided by subheadings. It should provide a concise and precise description of the experimental results, their interpretation, as well as the experimental conclusions that can be drawn." -this text must be deleted.

Response: Thank you very much for your comment and apologies for not having deleted the editorial guidelines on this section. We have deleted the respective text in lines 170-172.

Reviewer 2 Report

It is a remarkable paper about reducing the bitter-taste of pharmaceuticals using cell-based identification of bitter-masking compounds. It is well systematically written. No further revisions are required. 

Author Response

Comments provided by Reviewer #2

It is a remarkable paper about reducing the bitter-taste of pharmaceuticals using cell-based identification of bitter-masking compounds. It is well systematically written. No further revisions are required.

Response: Thank you for your comment! We very much appreciate that you find our work remarkable, and hope that our research will stimulate a new route for the development of pharmaceuticals with improved patients’ acceptance.

Reviewer 3 Report

     The manuscript submitted by Beltrán et al. reports on the in vitro detection of the effectiveness of bitter-taste masking compounds used in pharmaceutical applications, which were supported by human sensory trial.       While the abstract and introduction section of the manuscript were presented in rich details, I found the presentation of the experimental results not quite in line with the scope of the introduction. While the scientific background related the topic have been well-addressed in the Introduction section (especially between the lines 74-122), the presented background was still not helpful the follow the experimental results. From the very beginning of the results section, I have lost the scope of the hypothesis. I am not sure what was the actual aim of the research; whether it was presenting the relative effectiveness of the bitter-taste masking flavonoids or it was demonstrating an in vitro assaying method for the determination of such bitter-tasting masking activity by particularly using HGT-1 cells.     I have especially the following remarks, questions and suggestions about the experimental section, which requires the attention of Authors to improve the quality of presentation:   - SNARF-1 AM dye has been mentioned in abstract. However, there is nothing explained about the use of this dye in the introduction. Then, suddenly, the results section starts with the presentation of an experiment about the use of SNARF-1 AM that is really difficult to relate to the overall context of the manuscript. I think more explanation is needed about the role of SNARF-1 AM before the results section. - In lines 162-163, it is said: “finally, we compared both media in a 30 minutes-long measurement, where recordings under KRHP presented a more consistent profile over long time, …”. What does „over long time” statement relate to? Is there a experimental data presenting the data obtained in shorter time? I assumed that the Figure 2a was representing the data obtained in shorter time. However, I was not really able to relate Figure 2a with Figure 3. Anyways, if Figure 3 is presenting the results of 30 minutes-long experiment, then, what was the time period of experimental results presented in Figure 2a? Please clarify this point in details. - The results section 2.1 is highlighting the use if Krebs-HEPES buffer over DEMEM buffer. My first impression was that; the use of Krebs-HEPES buffer in proton-secretion assay of HGT-1 cells was a novel method. However, I realized that you have already used this method in your earlier paper published in Int. J. Mol. Sci. 2021, 22(11), 5881. However, I did not see any citation of the previous publication. I think, the citation must be included, and the hypothesis of the manuscript must be updated in the light of the previously published results. Presenting the use of Krebs-HEPES as if it was utilized for the first time along with HGT-1 cells does not provide a valid chronology of the scientific development of the topic.  - I am not sure what the statement in lines 170-172 actually relates to. Please provide further explanation. - In lines 185-186, “the marked changes” in HGT-1 cell surface were mentioned which were demonstrated in Figure S1. What type of changes were observed by the Authors? Could you please include detailed verbal description of such changes? Explanation of such changes from the Authors’ point of view would be much more explanatory for the readers.   I would like to particularly note that the discussion section is full of references from literature. Justification of the experimental results by providing too many references makes the text quite overwhelming to read. I believe, the manuscript must be improved by finding a good balance between the scientific background provided in introduction and discussion sections. I think an improved graphical presentation of the hypothesis, most likely by using a flow chart, could make it much easier for readers to understand the very tangled scientific background of the hypothesis. 

Author Response

General comment:

The manuscript submitted by Beltrán et al. reports on the in vitro detection of the effectiveness of bitter-taste masking compounds used in pharmaceutical applications, which were supported by human sensory trial.  While the abstract and introduction section of the manuscript were presented in rich details, I found the presentation of the experimental results not quite in line with the scope of the introduction. While the scientific background related the topic have been well-addressed in the Introduction section (especially between the lines 74-122), the presented background was still not helpful the follow the experimental results. From the very beginning of the results section, I have lost the scope of the hypothesis. I am not sure what was the actual aim of the research; whether it was presenting the relative effectiveness of the bitter-taste masking flavonoids or it was demonstrating an in vitro assaying method for the determination of such bitter-tasting masking activity by particularly using HGT-1 cells.  

Response: Thank you very much for your comment. We apologize for not having been clear enough. The actual aim of this work was to present an in vitro assaying method for the determination of bitter-tasting and bitter taste-masking pharmaceuticals. For this purpose, we optimized an already existing protocol using the human parietal cell line HGT-1. The suitability of this method was demonstrated by means of a known bitter-tasting drug, guaifenesin, and the reduction the guaifenesin-mediated bitter-response by addition of flavonoids as bitter-masking compounds. Finally, we verified our findings in a human sensory study.

That said, we revised the introduction of the manuscript in lines 141 – 146, text as follows:

“With this work, we present an in vitro assaying method for the determination of bitter-tasting and bitter taste-masking pharmaceuticals. For this purpose, we optimized an already existing protocol using human parietal cell line HGT-1 cells. The suitability of this method was demonstrated by means of a known bitter-tasting drug, guaifenesin, and the reduction the guaifenesin-mediated bitter-response by addition of flavonoids as bitter-masking compounds. Finally, we verified our findings in a human sensory study. Specifically, the previously published protocol – developed for food constituents – was optimized for its sensitivity by (i) removal of HCO3- through adaption of the cell culture medium, and (ii) the usage of inhibitors of solute carrier (SLC) transporters.“

I have especially the following remarks, questions and suggestions about the experimental section, which requires the attention of Authors to improve the quality of presentation:  

Comment 1: SNARF-1 AM dye has been mentioned in abstract. However, there is nothing explained about the use of this dye in the introduction. Then, suddenly, the results section starts with the presentation of an experiment about the use of SNARF-1 AM that is really difficult to relate to the overall context of the manuscript. I think more explanation is needed about the role of SNARF-1 AM before the results section.

Response: We also appreciate this comment, and tried to make the manuscript easier to understand for readers who are not familiar with this technique. We added the following part in lines 65 - 71: “One commonly used agent for this noninvasive measurement of the intracellular pH is carboxy-seminaphthorhodafluor-1 (carboxy-SNARF-1) [12]. This compound presents several advantages over classical fluorochromes such as 2,3-dicyano-hydrochinone (DCH) or 2,7-bis(2-carboxyethyl)-5(6)-carboxyfluorescein (BCECF), especially its low leakage and larger shift in its emission spectrum as a function of pH provides the ability to resolve small differences in pH over a longer period of time [13].”

Comment 2: In lines 162-163, it is said: “finally, we compared both media in a 30 minutes-long measurement, where recordings under KRHP presented a more consistent profile over long time, …”. What does „over long time” statement relate to? Is there a experimental data presenting the data obtained in shorter time? I assumed that the Figure 2a was representing the data obtained in shorter time. However, I was not really able to relate Figure 2a with Figure 3. Anyways, if Figure 3 is presenting the results of 30 minutes-long experiment, then, what was the time period of experimental results presented in Figure 2a? Please clarify this point in details.

Response: Thank you very much for your comment. In this context “over long time” refers to measurements over 30 min. Short times refer to measurements after 10 min of incubation. The measurement for 10 min was applied in our prior work (e.g. PNAS July 25, 2017 114 (30) E6260-E6269) [11]. In Figure 2a, the time period analyzed for the results presented was 10 minutes. This presentation is supplemented by the results shown in Figure 3, which present data for treatment times of up to 30 minutes – an approach that has not been published so far.

To clarify this point, the treatment times of 10 minutes have been added to the revised manuscript text as follows (lines 167 and 173).

“It was observed that cells under KRHP presented an up to ten times higher signal-to-noise ratio than their counterpart under DMEM after 10 minutes of treatment [10, 11] (Figure 2a).”

Figure 2. (a) Signal (SNARF-SNARF-1 AM loaded wells) to noise (blank wells) ratios of DMEM and KRHP media after 10 minutes of treatment.”

Comment 3: The results section 2.1 is highlighting the use if Krebs-HEPES buffer over DEMEM buffer. My first impression was that; the use of Krebs-HEPES buffer in proton-secretion assay of HGT-1 cells was a novel method. However, I realized that you have already used this method in your earlier paper published in Int. J. Mol. Sci. 2021, 22(11), 5881. However, I did not see any citation of the previous publication. I think, the citation must be included, and the hypothesis of the manuscript must be updated in the light of the previously published results. Presenting the use of Krebs-HEPES as if it was utilized for the first time along with HGT-1 cells does not provide a valid chronology of the scientific development of the topic. 

Response: Thank you very much for your comment and for having studied our previous work carefully. It is true that in our work published in Int. J. Mol. Sci. 2021, 22(11), 5881 we used Krebs-HEPES for the experiments to evaluate the serotonin release. In that work, the Krebs-HEPES buffer was chosen for the HGT-1 proton assay in order to use the same buffer as in the serotonin release assay. At that point, we had no knowledge about potentially different effects of DMEM compared to Krebs-HEPES buffer in the HGT-1 proton secretion assay. That is why we included this investigation in the here presented work. We absolutely agree that this point should be clarified and have added the following information to manuscript text (introduction, lines 128-131):

 “In one of our recent works, we used KREBS-Ringer HEPES buffer instead of the cell culture medium DMEM [20] in order to directly compare results from a serotonin-release assay with those of the proton secretion assay in HGT-1 cells.“

Comment 4: I am not sure what the statement in lines 170-172 actually relates to. Please provide further explanation.

Response: Thank you very much for your comment and apologies for not having deleted the editorial guidelines on this section. We have deleted the respective text on page 5, lines 170-172.

Comment 5: In lines 185-186, “the marked changes” in HGT-1 cell surface were mentioned which were demonstrated in Figure S1. What type of changes were observed by the Authors? Could you please include detailed verbal description of such changes? Explanation of such changes from the Authors’ point of view would be much more explanatory for the readers.  

Response: Thank you very much for your comment. We changed the text in lines 200-201 from “Under this treatment, marked changes in HGT-1 cell-surface are were observed (previously Figure S1, now Figure S2), indicating a higher amount of H+/K+-ATPases at the cell membrane level.” to “Under this treatment, marked changes in HGT-1 cell-surface were observed (Figure S2). In detail, a change in the morphology from the inactivated state to the stimulated status with an increased cell surface area is demonstrated. This indicates a higher amount of H+/K+-ATPases at the cell membrane level.”

Comment 6: I would like to particularly note that the discussion section is full of references from literature. Justification of the experimental results by providing too many references makes the text quite overwhelming to read. I believe, the manuscript must be improved by finding a good balance between the scientific background provided in introduction and discussion sections. I think an improved graphical presentation of the hypothesis, most likely by using a flow chart, could make it much easier for readers to understand the very tangled scientific background of the hypothesis. 

Response: Thank you very much for your comments. Regarding the amount of references, we would like to point out that we included a total number of 39 references, which is below the maximum amount recommended. When it comes to the balance between the scientific background and the discussion, we do agree with the reviewer that a higher number of references is included in the introduction. The reason for this is that this field of chemoreceptor-based research is relatively young. The link between ectopically expressed bitter receptor receptors and biological functions has been studied for no longer that about 15 years and is still in its infancy. In particular, knowledge on gastric functions related to the activation of chemoreceptors is relatively scarce. That is why the number of references included in the discussion is relatively low. However, we hope that our research presented here will stimulate research in that direction, as hypothesized in one of our recent papers entitled “Gastrointestinal taste receptors: could tastants become drugs?” by Behrens M, Somoza V. published in Curr Opin Endocrinol Diabetes Obes. 2020 Apr;27(2):110-114; doi: 10.1097/MED.0000000000000531.

To make the hypothesis easier to understand to the reader, we created the flow chart below. This chart has been added to the introduction as Figure S1. Since none of the other reviewers requested such an explanatory chart, we decided to include this Figure into the Supplemental Material in order to provide this information only to those readers who would like to get more details on the study design. In the manuscript text, we added the following information (lines 154-155): “A graphical overview of the experimental approach is shown in the supplementary Figure S1.”  

(for Figure S1: Please see file attached)

Figure S1: Experimental approach.
